

# Seagrass as major source of transparent exopolymer particles in the oligotrophic Mediterranean coast

Francesca Iuculano[1], Carlos M. Duarte[2], Núria Marbà[1], and Susana Agustí[2]

[1]Instituto Mediterráneo de Estudios Avanzados (IMEDEA), CSIC-UIB, Esporles, 07190, Balearic Islands, Spain.
[2] King Abdullah University of Science and Technology (KAUST), Red Sea Research Center (RSRC), Thuwal, 23955-6900, Saudi Arabia.

*Correspondence to*: Francesca Iuculano (fiuculano@imedea.uib-csic.es)





**Abstract.** The role of seagrass, *Posidonia oceanica*, meadows as a source of transparent exopolymer particles (TEP) to Mediterranean coastal waters was tested by comparing the TEP dynamics in two adjacent coastal waters in the oligotrophic

NW Mediterranean Sea, one characterized by oligotrophic open-sea waters and the other accumulating seagrass leaf litter, together with an experimental examination of TEP release by seagrass litter. TEP concentrations ranged from 4.6 µg XG Eq L$^{-1}$ to 90.6 µg XG Eq L$^{-1}$, with mean (± SE) values of 38.7 (± 2.02) µg XG Eq L$^{-1}$ in the site devoid of seagrass litter, whereas the coastal beach site accumulating leaf litter had > 10-fold mean TEP concentrations of 487.02 (± 72.8) µg XG Eq L$^{-1}$. Experimental evaluation confirmed high rates of TEP production by *P. oceanica* litter, allowing calculations of the

associated TEP yield. We demonstrated that *P. oceanica* is an important source of TEP to the Mediterranean Sea, contributing an estimated 0.10 Tg C as TEP annually. TEP release by *P. oceanica* seagrass explains the elevated TEP concentration relative to the low chlorophyll *a* concentration in the Mediterranean Sea.

## 1 Introduction

Transparent exopolymer particles (TEP) are acidic and sulphated polysaccharides enriched in deoxyc sugars and galactose

(Myklestad, 1977) which are stainable with Alcian Blue (Alldredge et al., 1993). These organic particles belong to the COC (colloidal organic carbon) pool (Zhou et al., 1998) and are ubiquitous in marine and limnetic ecosystems (Passow, 2002). Their roles in several biogeochemical processes and their importance in sedimentary carbon fluxes has been extensively documented (Engel and Passow, 2001) as, due to its sticky properties, the aggregation of these particles may enhance the sinking flux and export of organic matter (Kiørboe and Hansen, 1993; Simon et al., 2002). Phytoplanktonic cells, mainly

diatoms, are believed to be the major sources of TEP in the marine environment (Passow and Alldredge, 1995a), although benthic organisms, such as suspension feeders (Heinonen et al., 2007) and macroalgal detritus (Thornton, 2004) have been also identified as TEP sources. Indeed, marine macrophytes are important sources of dissolved organic carbon to coastal waters (Barron et al., 2006), and may therefore release precursors conducive to TEP formation, such as reported by Thornton (2004) for macroalgae. However, seagrass meadows are also important sources of DOC to the marine environment (Barrón

et al., 2014), but their role as a source of TEP has not yet been assessed.

*Posidonia oceanica* Delile (L.) is the dominant seagrass species of the Mediterranean Sea (Duarte, 2004). *P. oceanica* meadows are highly productive (Duarte and Chiscano, 1999) and release high amounts of dissolved organic carbon (Barron et al., 2014) as well as leaf litter (Cebrian and Duarte, 2001; Gacia et al., 2002). The large production of DOC and detritus by *P. oceanica* contrast with the low planktonic primary production in the oligotrophic Mediterranean littoral zone (Duarte et

al., 1999, Thingstad and Rassoulzadegan, 1999), where TEP are nevertheless present (Bar Zeev et al., 2011; Ortega et al., 2010; Prieto et al., 2006; Radić et al., 2005, Beauvais et al., 2003; Mari, 2001) at levels higher than expected as indicated by high TEP/Chl *a* and TEP/bacterial abundance ratios compared to other marine systems (Ortega et al., 2010). Whereas TEP are often assumed to be of phytoplankton origin, the relatively high levels of TEP (i.e. high TEP/Chl *a* ratios) in oligotrophic Mediterranean waters suggest that DOC release by *Posidonia oceanica* meadows could be a source of TEP, explaining the





relative high TEP concentration reported for Mediterranean waters (Ortega et al., 2010). Although macroalgae have been identified as sources of TEP, we are not yet aware of any studying examining the role of seagrass as a source of TEP.

In this study, we monitored the dynamics of TEP concentrations in two adjacent, but contrasting, oligotrophic littoral sites in Mallorca Island (NW Mediterranean Sea), an open coastline flushed with open sea waters and an adjacent, 2 Km, beach accumulated *Posidonia oceanica* leaf litter. We tested the hypothesis that seagrass leaf litter of *P. oceanica* represents an

important source of TEP to this ecosystem explaining the contrasting TEP concentrations and dynamics observed in these coastal sites using a laboratory experiment.

## 2 Materials and methods

### 2.1 Sampling sites and time series observations

The study was carried out at two sites in the coastal NW Mediterranean Sea off Majorca Balearic Island, a) the Faro Cap Ses

Salines experimental field station (Lat 39.265296 N; Lon 3.053427 E), where TEP concentration was monitored fortnightly for three years since 2006. This is a pristine and oligotrophic rocky shore ecosystem, with an extensive seagrass of *P. oceanica* meadow extended around 500 m offshore (Álvarez et al., 2015) and flushed with open-sea water (Fig. 1a), and b) Es Caragol beach (Lat 39.164285 N; Lon 3.23811 E), where TEP dynamics were monitored for two years. This is a natural sandy beach in a site of community importance (EU directive-red natura2000) where abundant seagrass detritus accumulates

in the shore (Fig. 1b), where it plays an important geomorphological role (Simeone and De Falco, 2012).

Surface water samples at Faro Cap Ses Salines and Es Caragol were collected fortnightly (monthly during winter months) on 2 L Nalgene bottles at noon and 3:00 pm, respectively. A total of 76 sampling events were completed at Faro Cap Ses Salines between 09 January 2012 to 23 March 2015, while 45 sampling events were completed at Es Caragol (from 09 August 2012 to 24 September 2014). Surface seawater samples of 250 mL from Faro Cap Ses Salines for chlorophyll *a*

determination were filtered through Whatman GF/F filters and stored at -20 ºC. Filters were extracted in 6 mL 90 % acetone for 24 hours followed by fluorometric (Trilogy, Turner designs) Chl *a* determination, calibrated with pure Chl *a*, after Pearsons et al. (1984). Sea-surface temperature was measured *in situ* using a data logger (HOBO).

TEP concentrations were determined following the colorimetric method of Passow and Alldredge (1995b), where TEP are detected after staining with Alcian Blue (Sigma), a cationic copper phthalocyanine dye that complexes carboxyl (-COO-)

and half-ester sulphate (OSO3-) reactive groups of acidic polysaccharides. Following each sampling event, triplicate aliquots (Faro Cap Ses Salines: 300-700 mL; Es Caragol: 50-500 mL, depending on the saturation of filters) were filtered onto 0.4 µm pore size, 25 mm diameter polycarbonate filters under low and constant pressure (150 mmHg). Filters were subsequently stained with 1000 µL of a 0.02 % working solution of Alcian Blue (pre-filtered through 0.2 µm) in 0.06 % acetic acid (pH = 2.5), allowed to stain for a few seconds, repeated filtering and rinsed twice with MilliQ water, to eliminate excess dye. Dyed

filters were stored at -80 ºC until extraction at IMEDEA laboratory. To perform the extraction, filters were placed in acid-clean 10 mL glass tubes, by adding 5 mL of 80 % sulphuric acid, for 2 to 3 hours, shaking 2 to 3 times to enhance extraction.





Absorbance was read spectrophotometrically (Shimadzu dual beam spectrophotometer) at 787 nm in 1 cm disposable cuvettes. Triplicate blank filters were also analysed for every batch of samples. Blank absorbance values at 787 nm were then subtracted from the total absorbance values of samples, to account for the capacity of Alcian Blue to stain filters. Eight

calibrations of the Alcian Blue solutions were performed by using Xanthan Gum as standard (XG). The calibration factor (F) was calculated as the mean of the eight estimates obtained. TEP concentrations (TEP) were expressed in µg Xanthan Gum (XG) equivalents per litre (µg XG Eq L$^{-1}$) and calculated following Eq. (1):

$$TEP = (a_{sample} - a_{blank})\ V^{-1} \cdot F, \tag{1}$$

where $a_{sample}$ and $a_{blank}$ are absorbance values at 787 nm for samples and blank filters, respectively; V is the sampled volume

(in L) and F is the calibration factor. The detection limit of the method was 2.2 µg XG Eq L$^{-1}$ and the analytical coefficient of variation was 13 %. TEP concentrations were transformed to carbon units (µg C L$^{-1}$) by using the conversion factor of 0.75 proposed by Engel and Passow (2001) in order to estimate the total TEP yield of *P. oceanica* leaf litter.

### 2.2 Experimental evaluation of TEP release by *P. oceanica* leaf litter

*P. oceanica* leaf litter and surface seawater were sampled on 8 September 2014, the time of leaf shedding for *P. oceanica,*

from the seashore of Es Caragol and stored at 4 ºC for transport to the laboratory. Six 5L Pyrex glass bottles were filled with seawater, pre-filtered by gravity through a 0.2 µm pore membrane size cartridge filter. Three replicated bottles received 16.6 mg fresh weight L$^{-1}$ of *P. oceanica* leaf litter, to obtain a final concentration similar to that measured in the near shore waters at Es Caragol, and three replicated bottles, without *P. oceanica* leaf litter, were used as control. The bottles were gently aerated with an air pump to provide mixing and avoid the development of anoxic conditions. The bottles were incubated at

the *in situ* temperature at the time of sampling (26.3 ºC) in a temperature controlled chamber, and water samples for TEP determinations were collected at increasing time intervals: time 0 (11 September), 6 hours, 12 hours, 12 hours, 48 hours and 264 hours (22 September) after the start of the experiment. The water volume and leaf biomass (fresh weight and dry weight following desiccation at 60 ºC for 24 hours in a drying oven) in the bottles were measured. Replicated 50 mL to 100 mL volumes, pre-filtered trough a 100 µm mesh to remove leaf litter, were sampled using a 60 mL syringe and immediately

filtered onto 0.4 µm to collect, dye and quantify TEP concentration following the procedure described above (Passow and Alldredge, 1995b).

### 3 Results

Surface seawater temperature ranged from 12.4 ºC to 27.8 ºC, registered in February 2012 and September 2014, respectively, along the study (average ± SE = 19.4 ± 0.54 ºC). Chlorophyll *a* concentration ranged from 0.02 to 0.54 µg L$^{-1}$ in July 2014

and March 2013, respectively, along the study (average ± SE = 0.23 ± 0.01 µg L$^{-1}$).

TEP concentrations ranged from 4.6 to 90.6 µg XG Eq L$^{-1}$ in Faro Cap Ses Salines and from 26.8 to 1878.4 µg XG Eq L$^{-1}$ in Es Caragol, with significantly (paired t-test, P < 0.05) higher mean TEP concentrations at Es Caragol (38.7 ± 2.02 µg XG Eq L$^{-1}$) compared to Faro Cap Ses Salines (487.02 ± 72.8 µg XG Eq L$^{-1}$). TEP concentrations changed greatly seasonally, with

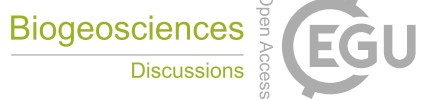



maximum TEP values in waters sampled at the Faro Cap Ses Salines observed in February, likely associated with the phytoplankton bloom occurring at that time, and June (Fig. 2a). In contrast, TEP dynamics showed a more erratic temporal pattern at Es Caragol, with no clear seasonal patters (Fig. 2b). Mean (± SE) TEP/Chl *a* ratios were also > 10-fold greater at Es Caragol (3109.9 ± 468.9) than at the Faro Cap Ses Salines (286.3 ± 55.7), with a clear seasonal cycle characterized by maximum TEP/Chl *a* ratios in June and July at the Faro Cap Ses Salines whereas at Es Caragol they remained elevated throughout the year, except between January and March when values were relatively low (Fig. 3a, b).

During the experimental evaluation initial TEP concentrations (30.4 µg XG Eq $L^{-1}$) increased slightly after 6 h incubation, to remain uniform throughout the rest of the experiment in the absence of *P. oceanica* leaf litter (Fig. 4). In contrast, TEP concentrations increased greatly throughout the experiment in the presence of *P. oceanica* litter, reaching values of 1551 µg XG Eq $L^{-1}$, comparable to maximum values observed at Es Caragol, after 264 h (Fig. 4). The corresponding TEP yield of *P. oceanica* corresponded to 14.128 ± 11.294 µg XG Eq $L^{-1}$ or 2344 ± 357.26 µg C g $DW^{-1}$. The yield of TEP in the presence of *P. oceanica* litter was 9.77 times greater than that in control bottles (1.384 ± 1.582 µg XG Eq $L^{-1}$).

## 4 Discussion

The results presented provide, to the best of our knowledge, the first evidence that seagrass leaf litter is a source of TEP to coastal waters. Thornton (2004) demonstrated the formation of TEP from the acidic polysaccharides released by macroalgal detritus of different species, but the role of seagrass litter as a source of TEP has not been reported to-date. The role of *P. oceanica* leaf litter as a source of TEP is demonstrated here through the > 10-fold difference in concentration and TEP/Chl *a* ratios between the two adjacent coastal areas studied, one containing rapidly flushed open-sea water and the other representing an accumulation site for *P. oceanica* leaf litter. The experimental evidence reported further confirms the role of TEP formed by precursors released by *P. oceanica* leaf litter in explaining the differences between the two sites, as the TEP concentration reached, using a concentration of leaf litter similar to that observed in Es Caragol, is comparable to the maximum values observed *in situ*.

*P. oceanica*, as well as seagrasses in general, exports a large fraction of its net primary production as leaf litter, on average about 24 % of NPP (Duarte and Cebrian, 1996). A fraction of this leaf litter is exported to the shoreline following leaf shedding by *P. oceanica* in the late summer and early autumn (Mateo et al., 2003). Leaf litter is then deposited on the beach and re-entrained in the water during storms, resulting in the pulses of TEP observed at Es Caragol.

The seasonal variability in TEP/Chl *a* ratios at Faro Cap Salines, where leaf litter accumulation is precluded by strong currents, shows a maximum in the summer (June and July), likely resulting from TEP precursors released by the nearby seagrass meadow. Ortega et al. (2010) already reported elevated TEP/Chl *a* ratios during early summer in the Mediterranean Sea, with values comparable to those we observe at the Faro Cap Salines. These observations suggest that *P. oceanica* meadows, the dominant ecosystem in Mediterranean coastal waters, are the source of TEP precursors responsible for the elevated TEP/Chl *a* ratios characteristic of the Mediterranean Sea (Ortega et al., 2010).





Considering the average production of *P. oceanica* of 2.40 g DW m$^{-2}$ d$^{-1}$ (Duarte and Chiscano, 1999), the estimated 50,000 Km$^2$ covered by *P. oceanica* in the Mediterranean (Bethoux and Copin-Montegut, 1986) and the average TEP yield from leaf litter experimentally derived here (2344 µg C g DW$^{-1}$) we calculated that *P. oceanica* releases about 0.10 Tg C as TEP annually to the Mediterranean Sea. This estimate highlights the important role of seagrass litter as source of TEP in the

Mediterranean, a role that may play seagrasses in other regions with lush seagrass meadows, as the Caribbean or South East Asia.

Seagrass meadows have been recently shown to be globally relevant sources of DOC to the marine ecosystem (Barron et al., 2014) and Mari et al., 2017 have recently assess that the global TEP production could represent 2.5 to 5 Pg C y$^{-1}$. Here we provide the first evidence that seagrass meadows can also play an important, even locally dominant, role as sources of TEP

and, therefore, particle dynamics in the ocean. This finding has important biogeochemical implications and provides a new pathway to be accounted for when considering the fate and fluxes of organic matter in the continuum of DOM-POM bridge.

*Competing interests*. The authors declare that they have no conflict of interest.

*Acknowledgments*. This work is a contribution to the StressX project, funded by the Spanish Ministry of Economy and Innovation (CTM2012-32603). F.I. was supported by JAE predoctoral fellowship from the Consejo Superior de

Investigaciones Científicas (CSIC). We thank J. C. Martinez for help with sampling and Chl *a* measurements.

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




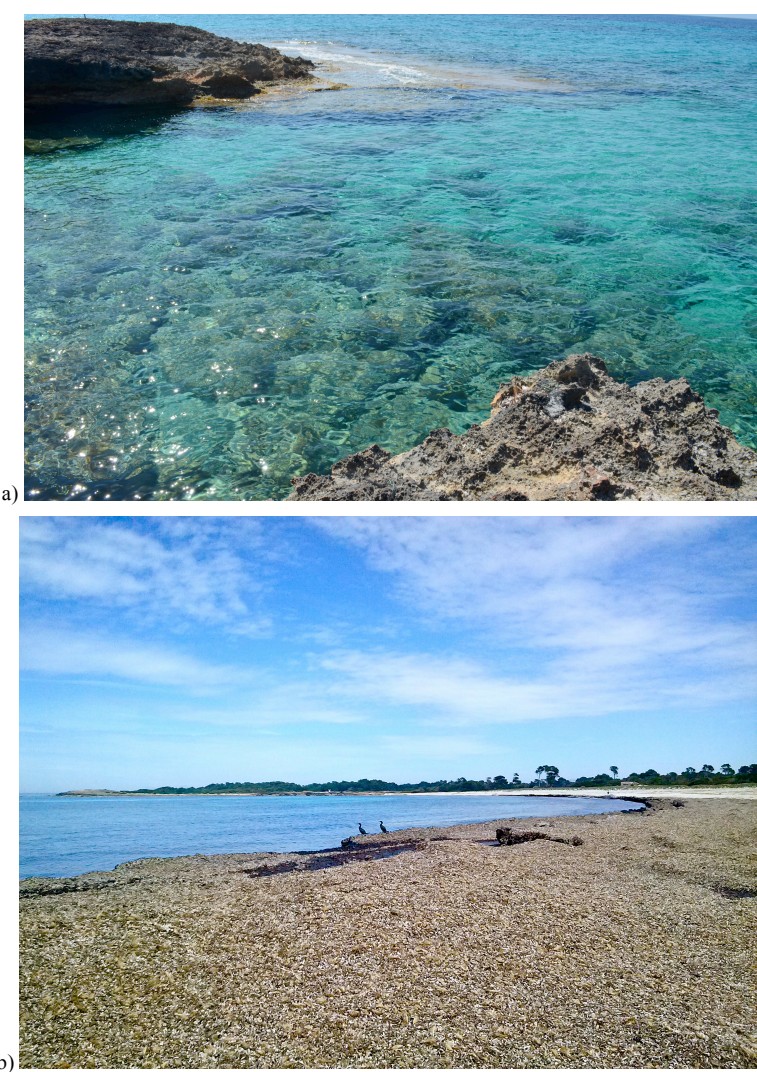

**Figure 1: The two sites monitored at Cap Ses Salines (a) and at Es Caragol beach (b), Mallorca island, NW Mediterranean.**





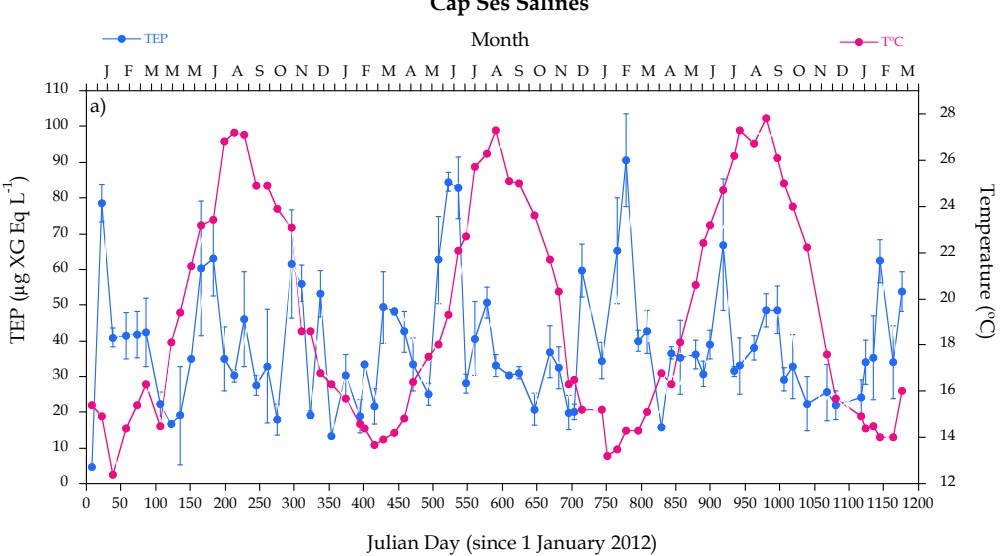

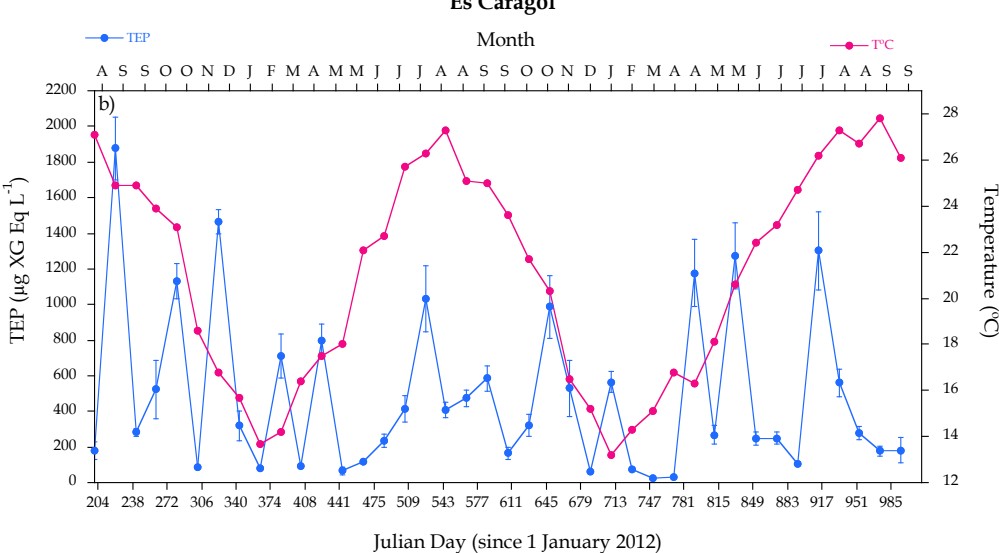


**Figure 2: Time series of TEP concentrations (µg XG Eq L$^{-1}$ ± SE) and Temperature (ºC) at Cap Ses Salines (a) and Es Caragol (b).**





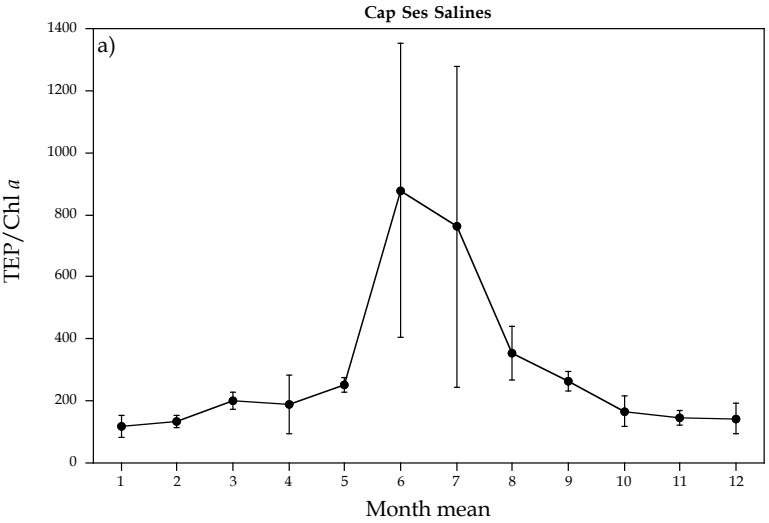

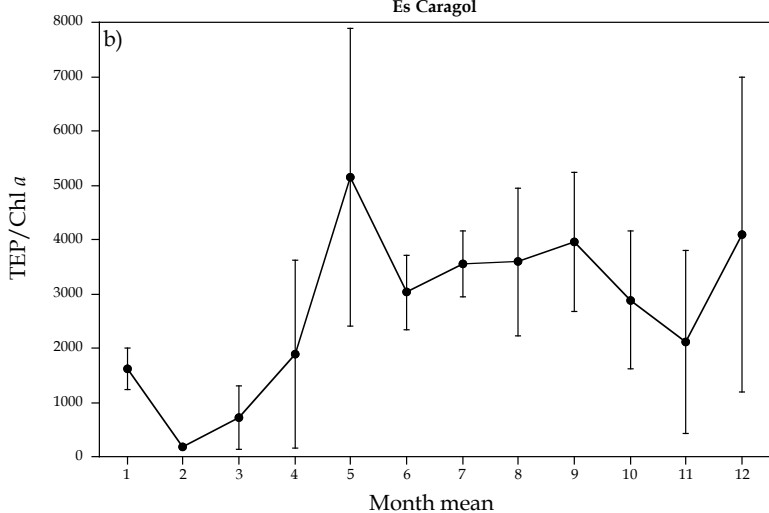

**Figure 3: Monthly TEP/Chl *a* ratios means ± SE at Cap Ses Salines (a) and Es Caragol (b).**



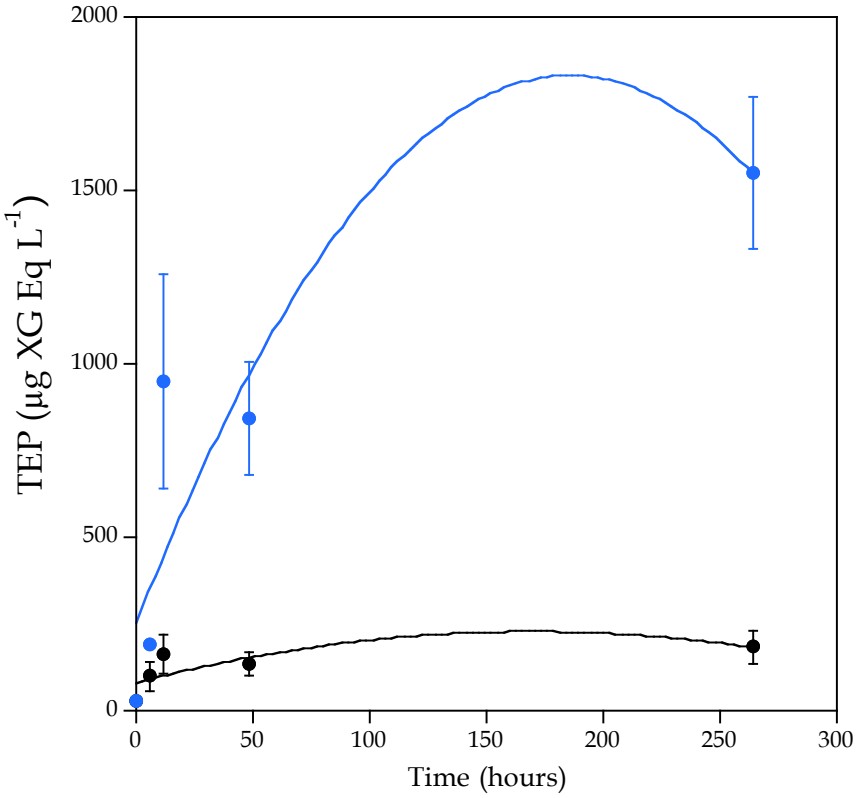

**Figure 4: TEP accumulation (mean ± SE) in the presence (blue line) and absence (dark line) of *P. oceanica* litter. The solid lines show the fitted second order polynomial equations ($R^2$= 0.77 and 0.53, respectively).**
