# Peer review of "Seagrass as major source of transparent exopolymer particles in the"

_Biogeosciences, 2016_

## Referee Comment (RC1) · Anonymous Referee #1 · 3 Mar 2017

General comments.

This paper focuses on the production of TEP by the litter of Posidonia oceanica, and discusses the implications of such TEP production in the context of the oligotrophic Mediterranean Sea. This work combines a field study conducted for 2-3 years in two coastal sites with a laboratory experiment. The first field study site (Faro Cap Ses Salines) is a pristine and oligotrophic rocky shore system, and the second one (Es Caragol beach) is a sandy beach covered with abundant seagrass detritus. TEP dynamics were monitored at these two sites and were compared in order to determine the impact of seagrass detritus on TEP concentration. This field study was completed by a laboratory study during which P. oceanica leaf litter collected from the seashore of

[Figure]

Es Caragol beach was incubated at 26.3°C in seawater collected at the same site, and TEP production was monitored during 264 hours.

This work is interesting because it is the first to report the impact of P. oceanica leaf litter on TEP dynamics in a coastal site. This work clearly shows that the presence of leaf litter is at the origin of a significant TEP production. This conclusion is drawn from the fact that TEP concentrations were on average 10-fold higher in the coastal beach accumulating leaf litter, in comparison to TEP concentrations in the rocky shore system. The high production of TEP from decomposing leaf litter in the laboratory confirms the field observation.

While this work shows convincingly that P. oceanica detritus can be at the origin of a high TEP production in coastal areas of the oligotrophic Mediterranean Sea, the discussion concerning the impact of such TEP production is less convincing.

Specific comments.

The main issue concerns the conclusions drawn from the results obtained, i.e. sea-grass is a major source of TEP that can explain the elevated TEP concentration relative to the low chlorophyll a concentration in the Mediterranean Sea. This statement comes from the estimate of TEP production from seagrass obtained, 0.10 Tg C per year, which in my opinion is an absolute maximum for the following reasons:

- This estimate of the release rate of TEP from decomposing seagrass was obtained using the highest values of the surface area covered by P. oceanica in the Mediterranean Sea (50,000 km2), while estimates are from 25,000 to 50,000 km2 (e.g. Borum et al. 2004, Boudouresque et al. 2006).

- This estimate is based on the assumption that the TEP yield from leaf litter in the Mediterranean Sea is constant during the year and equals that measured in the laboratory under controlled conditions, i.e. 2344 $\mu$g TEP-C DW-1. But this estimate was obtained by incubating leaf litter at 26.3°C, while this temperature is not met all over

the year in the Mediterranean Sea (reported range in the present work from 12.4 to 27.8°C), and that this decomposition process is temperature dependant (e.g. Øster-gaard Pedersen et al. 2011). The latter study shows that the maximum decomposition rate occurs around 25°C. In addition, the reported temperature range from 12.4 to 27.8°C is for surface waters, while a large fraction of the P oceanica leaf is exported in deep waters (i.e. at temperature lower than in surface waters). Therefore, the estimate given assumes that all the leaf litter is decomposed at 26.3°C, although even when the surface temperature is 26.3°C, a significant fraction of leaf decomposes in lower temperature, at depth.

- This estimate is based on an average production of P. oceanica of 2.40 g DW m-2 d-1 (i.e. 876 g DW m-2 y-1). While this estimate of P. oceanica production seems correct for the whole plant, the leaf production is much lower, and varies between 50 and 150 g DW m-2 y-1 (Pergent et al. 1994, 1997). In my opinion, the leaf production should be used since it is at the origin of leaf litter.

The estimate of the TEP production from seagrass (leaf litter) should acknowledge the above limitations, and a range of production should be given. Providing only a maximum theoretical value as a reference value can lead to a severe overestimation of the role of seagrass in TEP production in the Mediterranean Sea.

In addition, it should be noted that the elevated TEP concentration relative to the low chlorophyll a concentration in the Mediterranean Sea does not need TEP production from seagrass to be explained. It has been showed that the relationship between TEP and chlorophyll a concentrations was only linear during the growth phase of phyto-plankton blooms, and that TEP could accumulate in the water column of a oceanic site (no impact of leaf litter expected) in the Mediterranean Sea at low chlorophyll a concentration, due to carbon overflow during nutrient limiting conditions (e.g. Mari et al. 2001, Radić et al. 2005). This leads to a summer accumulation of TEP in surface layer of the stratified Mediterranean Sea at low chlorophyll concentration, explaining the high TEP/chloro during this period (as shown in Figure 3). Therefore, I think the

last sentence on page 5 (lines 133-135) "These observations suggest that P. oceanica meadows, the dominant ecosystem in Mediterranean coastal waters, are the source of TEP precursors responsible for the elevated TEP/Chl a ratios characteristic of the Mediterranean Sea" is not correct.

Lines 123-125: This sentence suggests that TEP precursors are released directly by P. oceanica leaf litter. How about the microbial activity associated with litter? The role of microbial communities associated with leaf litter should be acknowledged.

Technical corrections.

Line 35: The work by Thingstad and Rassoulzadegan (1999) does not discuss the littoral zone of the Mediterranean Sea.

Lines 33-40: This paragraph is not logical as it is based on a pivotal argument that is not valid. It says (1) P. oceanica produces high amounts of detritus, (2) in an oligotrophic environment, (3) oligotrophic, but with TEP concentration exceeding what can be expected from TEP/chl a or TEP/bact abundance ratios, (4) this discrepancy can only be explained by a high TEP production by P. oceanica. But as explained above, there is no linear relationship between TEP and chlorophyll concentrations.

Line 41: Study instead of studying.

References cited.

Borum J, Duarte CM, Greve TM, Krause-Jensen D (Eds.) (2004) European seagrasses: an introduction to monitoring and management (p. 2006). M&MS project. Boudouresque CF, Bernard G, Bonhomme P, Charbonnel E, Diviacco G, Meinesz A, Pergent G, Pergent-Martini C, Ruitton S, Tunesi L (2006) Préservation et conservation des herbiers à Posidonia oceanica. ramoge publ. (ISBN 2-905540-30-3), Monaco: 200 pp. Østergaard Pedersen M, Serrano O, Mateo MA, Holmer M (2011) Temperature effects on decomposition of a Posidonia oceanica mat. Aquatic Microbial Ecology, 65: 169–182. Pergent G, Romero J, Pergent-Martini C, Mateo MA, Boudouresque

[Figure]

CF (1994) Primary production, stocks and fluxes in the Mediterranean seagrass Posidonia oceanica. Marine Ecology Progress Series, 106: 139–146. Pergent G, Rico-Raimondino V, Pergent-Martini C (1997) Fate of primary production in Posidonia oceanica meadows of the Mediterranean. Aquatic Botany, 59: 307–321. Radić T, Kraus R, Fuks D, Radić J, Pečar O (2005) Transparent exopolymeric particles' distribution in the northern Adriatic and their relation to microphytoplankton biomass and composition. Science of the Total Environment, 353(1): 151-161.
* * *

---

## Referee Comment (RC2) · Anonymous Referee #2 · 16 Jun 2017

**Seagrass as major source of transparent exopolymer particles in the oligotrophic Mediterranean coast**

This paper by Iuculano et al. evaluates the seagrass Posidonia oceanica as a source of TEP to Mediterranean coastal waters by comparing two sites, an oligotrophic rocky shore (Faro Cap Ses Salines) and a sandy beach with seagrass detrital accumulation (Es Caragol). Additionally, the authors experimentally quantified release of TEP by P. oceanica leaf litter in the laboratory over the course of 11 days. The study is interesting and seems to clearly implicate P. oceanica as a source of TEP locally based on the significantly higher amounts of TEP at Es Caragol compared to Faro

Cap Ses Salines, and the production of TEP from the leaf litter in the lab experiment is evident. Expanding these results out to the whole of the Mediterranean needs to be done with a bit more caution, however. It should be more clearly indicated that these estimates would likely be maximum values of TEP contribution by P. oceanica, as the value used for the calculations came from the lab experiments conducted at 26.3°C which is near the maximum annual temperature at Es Caragol. The fate of this TEP is also not addressed within the scope of the study, so again its persistence or overall contribution to the entire Med Sea is not clear as these hyper-local source sites may be incredibly patchy and behave very differently in different locations. Yet these results are still an useful starting point for quantifying TEP from seagrasses, and so I recommend publication if the limitations of the concluding estimates are addressed in the text.

**Minor comments:**
- line 19: it appears there's a typo here, should 'deoxyc' be 'deoxy sugars'?
- line 41: 'studying' should be 'study'
- line 51: 'for three years since 2006' is confusing here, especially since the data in figure 2 start in January 2012? please clarify
- line 55: 'in the shore' should be changed to 'on the shore'
- line 56: 'on 2 L Nalgene bottles' should be changed to 'in 2 L Nalgene bottles'
- line 91: 12 hours is listed twice here, please correct
- line 140: 'a role that may play seagrasses' is confusing and appears out of order, please clarify
- line 143: 'assess' should be 'assessed' here
- line 145: the wording of the end of this final sentence is awkward and should be revised
- Bar Zeev et al., 2011, Duarte and Cebrian, 1996, and Myklestad, 1977 are not listed in the references.
- Myklestad, 1995 and Parsons et al., 1984 are listed in the references but not referenced in the text.

---

## Author Comment (AC1) · 11 Aug 2017

Answer to the general comment:

We thank the reviewer for the assessment that our results clearly show the response of TEP by Posidonia oceanica leaf litter release in the coastal area of the oligotrophic Mediterranean Sea and our field observations are confirmed by the experiment conducted in the laboratory. We also agree with the reviewer that, as also pointed out by reviewer 1, the discussion on the role of TEP release by P. oceanica at the scale of the Mediterranean basin needs be improved by acknowledging uncertainties around the estimates provided.

[Figure]

We will revise the discussion to read:

"These observations suggest that P. oceanica meadows, the dominant ecosystem in Mediterranean coastal waters, are an important source of TEP precursors in the Mediterranean Sea. Considering the average leaf production of P. oceanica of 876 g DW m-2 y-1 (Duarte and Chiscano, 1999), the estimated 37,000 Km2 covered by P. oceanica in the Mediterranean Sea (range 31,040 to 43,550 Km2, Marbá et al. 2014), and the average TEP yield from leaf litter experimentally derived here (2344 $\mu$g C g DW-1) we calculated that P. oceanica releases about 76 Gg C as TEP annually to the Mediterranean Sea. However, this estimate should be considered a first-order estimate, as it involves considerable uncertainty, compounding that derived from the substantial variability in primary production of P. oceanica (Duarte and Chiscano, 1999), that in the area covered by P. oceanica meadows in the Mediterranean Sea, and variability in TEP yield across meadows and over time, as the estimate used was derived from a single meadow in the fall. Improving this estimate will require narrowing down these sources of uncertainty as well as the capacity to compare it with estimates of other sources of TEP, such as phytoplankton, which are not yet available at the basin scale. The contribution of P. oceanica meadows to TEP release may contribute to explain, along with other processes, the elevated TEP/Chl a ratios characteristic of the Mediterranean Sea (Ortega et al., 2010). The role of P. oceanica as a relevant source of TEP precursors is enhanced by the contrast between the high production of P. oceanica meadows (Duarte and Chiscano, 1999), resulting in a high production of detritus (e.g. Mateo and Romero 1997, Cebrian and Duarte 2001) releasing TEP precursors, and the oligotrophic nature of the Mediterranean Sea, leading to low production in the pelagic compartment. In fact, both P. oceanica (e.g. Alcoverro et al., 1997) and phytoplankton (e.g. Krom et al., 1991) are likely to be strongly nutrient-limited in the Mediterranean Sea, which has been shown to enhance the release of TEP precursors through carbon overflow during nutrient limiting conditions (Mari et al., 2001; Radic et al., 2005). Despite the limitations acknowledge above, our estimates highlight the important role of P. oceanica litter as source of TEP in the Mediterranean, and suggest

that seagrass meadows may play a similarly important role in other regions supporting extensive seagrass meadows, such as the Caribbean, Australia and South East Asia."

Answer to the minor comments:

Line 19: We agree. The revised version of the manuscript will correct "deoxyc" with "deoxy".

Line 41: We agree. The revised version of the manuscript will correct "studying" with "study".

Line 51: We agree. The revised version of the manuscript will clarify "for three years since 2006" with "for three years since January 2012". (This study started in January 2012 in Cap Ses Salines and in August 2012 in Es Caragol beach. However, the time series project in Cap Ses Salines started in 2006. We agree that it is not necessary give this detail as it may confound the reader). We will also add in line 54 "for two years since August 2012", when sampling in Es Caragol beach started.

Line 55: We agree. The revised version of the manuscript will correct "in the shore" with "on the shore".

Line 56: We agree. The revised version of the manuscript will correct "on 2 L Nalgene bottles" with "in 2 L Nalgene bottles".

Line 91: "12 hours" listed twice in the revised version of the manuscript will be corrected with "24 hours" as we also sampled at this time interval.

Line 140: We agree, the revised version will be corrected to read: "Despite the limitations acknowledge above, our estimates highlight the important role of P. oceanica litter as source of TEP in the Mediterranean, and suggest that seagrass meadows may play a similarly important role in other regions supporting extensive seagrass meadows, such as the Caribbean, Australia and South East Asia".

Line 143: We agree. The revised version of the manuscript will correct "assess" with

"assessed".

Line 145: We agree. The revised version of the manuscript will add "for the" particles dynamics in the ocean at the end of this sentence.

Bar Zeev et al., 2011; Duarte and Cebrian, 1996 will be cited in the revised version of the manuscript. Myklestad, 1977 will be changed with Myklestad, 1995 in the text line 20 and in the reference list.

Parsons et al., 1984 yes it is already cited in the text in line 63.

Please also note the supplement to this comment:
https://www.biogeosciences-discuss.net/bg-2016-558/bg-2016-558-AC1-
supplement.pdf

―――――――――――――――――――

---

## Author Comment (AC2) · 11 Aug 2017

Answer to the general comments:

We thank the reviewer for the positive assessment acknowledging the research reported as original and interesting. We also thank the reviewer for the comments and suggestions for improvements, which will help produce a much improved revised version of the manuscript. We will improve the manuscript following the constructive reviews and adding the references proposed (see specific actions below).

We agree that the discussion on the impact of TEP production by P. oceanica at the

scale of the Mediterranean suffers from a number of limitations, including a number of uncertainties around the TEP production at the basin scale, which derive from uncertainties in the components of such estimate. We will address these limitations and will, on the light of these limitations, tone down the claims at the scale of the Mediterranean. We will also revise the text to improve clarity.

This text will be changed in the revised version to:

"These observations suggest that P. oceanica meadows, the dominant ecosystem in Mediterranean coastal waters, are an important source of TEP precursors in the Mediterranean Sea. Considering the average leaf production of P. oceanica of 876 g DW m-2 y-1 (Duarte and Chiscano, 1999), the estimated 37,000 Km2 covered by P. oceanica in the Mediterranean Sea (range 31,040 to 43,550 Km2, Marbà et al. 2014), and the average TEP yield from leaf litter experimentally derived here (2344 $\mu$g C g DW-1) we calculated that P. oceanica releases about 76 Gg C as TEP annually to the Mediterranean Sea. However, this estimate should be considered a first-order estimate, as it involves considerable uncertainty, compounding that derived from the substantial variability in primary production of P. oceanica (Duarte and Chiscano, 1999), that in the area covered by P. oceanica meadows in the Mediterranean Sea, and variability in TEP yield across meadows and over time, as the estimate used was derived from a single meadow in the fall. Improving this estimate will require narrowing down these sources of uncertainty as well as the capacity to compare it with estimates of other sources of TEP, such as phytoplankton, which are not yet available at the basin scale. The contribution of P. oceanica meadows to TEP release may contribute to explain, along with other processes, the elevated TEP/Chl a ratios characteristic of the Mediterranean Sea (Ortega et al., 2010). The role of P. oceanica as a relevant source of TEP precursors is enhanced by the contrast between the high production of P. oceanica meadows (Duarte and Chiscano, 1999), resulting in a high production of detritus (e.g. Mateo and Romero, 1997; Cebrian and Duarte, 2001) releasing TEP precursors, and the oligotrophic nature of the Mediterranean Sea, leading to low production

in the pelagic compartment. In fact, both P. oceanica (e.g. Alcoverro et al., 1997) and phytoplankton (e.g. Krom et al., 1991) are likely to be strongly nutrient-limited in the Mediterranean Sea, which has been shown to enhance the release of TEP precursors through carbon overflow during nutrient limiting conditions (Mari et al., 2001; Radić et al., 2005). Despite the limitations acknowledge above, our estimates highlight the important role of P. oceanica litter as source of TEP in the Mediterranean, and suggest that seagrass meadows may play a similarly important role in other regions supporting extensive seagrass meadows, such as the Caribbean, Australia and South East Asia".

We provide below, our response to the specific comments:

We agree that the estimate of the TEP production of 0.10 Tg C y-1 we propose involve significant uncertainties, which should be acknowledged, and represent, therefore, a first-order estimate. We will acknowledge and discuss these limitations in the revised version of the manuscript (see above). We agree that the area covered involves uncertainties, and now report the range of the most robust assessment to date (range 31,040 to 43,550 Km2, Marbà et al. 2014). We will revise the estimate of TEP production, accordingly, downwards to 76 Gg C y-1 (compared to 0.1 Tg C reported originally), and acknowledge that this is a first-order estimate with substantial uncertainty. We will also acknowledge that the yield of TEP may be variable across meadows and over time, and that these variability need be considered (see above). The production reported above of 876 g DW m-2 y-1 (Duarte and Chiscano, 1999), is indeed leaf production, and is a more thorough estimate, the average of 17 estimates, than those provided in Pergent et al. (1994, 1997). We also acknowledge that this estimate carries significant uncertainty (reported in Duarte and Chiscano, 1999).

Line 123-125: We will acknowledge that the leaf litter of P. oceanica supports a complex community of heterotrophic microbes that may contribute to TEP release. The text will be revised to read: "The experimental evidence reported further confirms the role of TEP formed by precursors released by P. oceanica leaf litter, together with the associated microbial heterotrophic community (Peduzzi et al. 1991), in explaining the

differences between the two sites, as the TEP concentration reached, using a concentration of leaf litter similar to that observed in Es Caragol, is comparable to the maximum values observed in situ."

Answer to the technical corrections:

Line 35: We agree, in the new version of the manuscript we deleted the reference of Thingstad and Rassoulzadegan (1999).

Line 33-40: We agree and the text has been modified accordingly (see above).

Line 41: We agree, the new version of the manuscript will be corrected from "study" to "studying".

The following references will be included in the revised manuscript:

Alcoverro, T., Romero, J., Duarte, C. M. and López, N. I.: Spatial and temporal variations in nutrient limitation of seagrass Posidonia oceanica growth in the NW Mediterranean, Mar. Ecol. Prog. Ser., 146, 155–161, 1997.

Cebrian, J. and Duarte, C. M.: Detrital stocks and dynamics of the seagrass Posidonia oceanica (L.) Delile in the Spanish Mediterranean, Aquat. Bot., 70, 295–309, doi:10.1016/S0304-3770(01)00154-1, 2001.

Krom, M. D., Kress, N., Brenner, S. and Gordon, L. I.: Phosphorus limitation of primary productivity in the eastern Mediterranean Sea, Limnol. Oceanogr., 36(3), 424–432, 1991.

Marbà, N., Díaz-Almela, E. and Duarte, C. M.: Mediterranean seagrass (Posidonia oceanica) loss between 1842 and 2009, Biol. Conserv., 176, 183–190, doi:10.1016/j.biocon.2014.05.024, 2014. Mateo, M. A. and Romero, J.: Detritus dynamics in the seagrass Posidonia oceanica: elements for an ecosystem carbon and nutrient budget, Mar. Ecol. Prog. Ser., 151, 43–53, 1997.

Peduzzi, P. and Herndl, G. J.: Decomposition and significance of seagrass leaf litter

(Cymodocea nodosa) for the microbial food web in coastal waters (Gulf of Trieste, Northern Adriatic Sea), Mar. Ecol. Prog. Ser., 71, 163–174, 1991.

Please also note the supplement to this comment:
https://www.biogeosciences-discuss.net/bg-2016-558/bg-2016-558-AC2-supplement.pdf